# Denoising Diffusion Probabilistic Models

**Jonathan Ho**
UC Berkeley
jonathanho@berkeley.edu

**Ajay Jain**
UC Berkeley
ajayj@berkeley.edu

**Pieter Abbeel**
UC Berkeley
pabbeel@cs.berkeley.edu

## Abstract

We present high quality image synthesis results using diffusion probabilistic models, a class of latent variable models inspired by considerations from nonequilibrium thermodynamics. Our best results are obtained by training on a weighted variational bound designed according to a novel connection between diffusion probabilistic models and denoising score matching with Langevin dynamics, and our models naturally admit a progressive lossy decompression scheme that can be interpreted as a generalization of autoregressive decoding. On the unconditional CIFAR10 dataset, we obtain an Inception score of 9.46 and a state-of-the-art FID score of 3.17. On 256x256 LSUN, we obtain sample quality similar to ProgressiveGAN. Our implementation is available at https://github.com/hojonathanho/diffusion.

## 1 Introduction

Deep generative models of all kinds have recently exhibited high quality samples in a wide variety of data modalities. Generative adversarial networks (GANs), autoregressive models, flows, and variational autoencoders (VAEs) have synthesized striking image and audio samples [14, 27, 3, 58, 38, 25, 10, 32, 44, 57, 26, 33, 45], and there have been remarkable advances in energy-based modeling and score matching that have produced images comparable to those of GANs [11, 55].

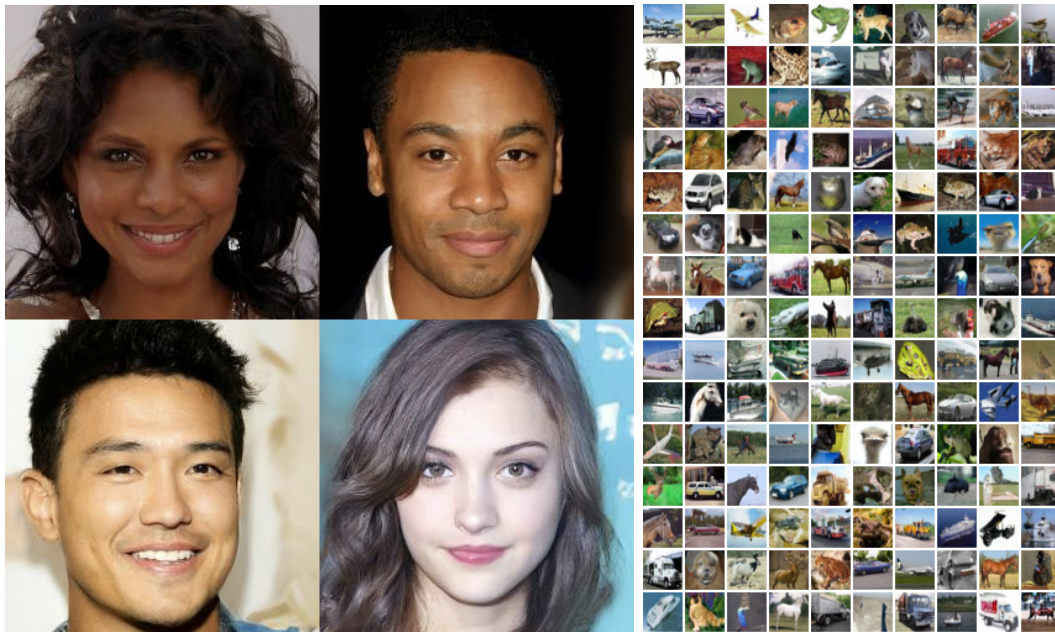

Figure 1: Generated samples on CelebA-HQ $256 \times 256$ (left) and unconditional CIFAR10 (right)

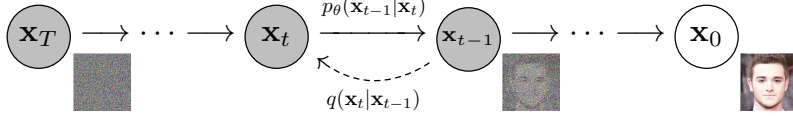

Figure 2: The directed graphical model considered in this work.

This paper presents progress in diffusion probabilistic models [53]. A diffusion probabilistic model (which we will call a "diffusion model" for brevity) is a parameterized Markov chain trained using variational inference to produce samples matching the data after finite time. Transitions of this chain are learned to reverse a diffusion process, which is a Markov chain that gradually adds noise to the data in the opposite direction of sampling until signal is destroyed. When the diffusion consists of small amounts of Gaussian noise, it is sufficient to set the sampling chain transitions to conditional Gaussians too, allowing for a particularly simple neural network parameterization.

Diffusion models are straightforward to define and efficient to train, but to the best of our knowledge, there has been no demonstration that they are capable of generating high quality samples. We show that diffusion models actually are capable of generating high quality samples, sometimes better than the published results on other types of generative models (Section 4). In addition, we show that a certain parameterization of diffusion models reveals an equivalence with denoising score matching over multiple noise levels during training and with annealed Langevin dynamics during sampling (Section 3.2) [55, 61]. We obtained our best sample quality results using this parameterization (Section 4.2), so we consider this equivalence to be one of our primary contributions.

Despite their sample quality, our models do not have competitive log likelihoods compared to other likelihood-based models (our models do, however, have log likelihoods better than the large estimates annealed importance sampling has been reported to produce for energy based models and score matching [11, 55]). We find that the majority of our models' lossless codelengths are consumed to describe imperceptible image details (Section 4.3). We present a more refined analysis of this phenomenon in the language of lossy compression, and we show that the sampling procedure of diffusion models is a type of progressive decoding that resembles autoregressive decoding along a bit ordering that vastly generalizes what is normally possible with autoregressive models.

## 2   Background

Diffusion models [53] are latent variable models of the form $p_\theta(\mathbf{x}_0) \coloneqq \int p_\theta(\mathbf{x}_{0:T}) \, d\mathbf{x}_{1:T}$, where $\mathbf{x}_1, \ldots, \mathbf{x}_T$ are latents of the same dimensionality as the data $\mathbf{x}_0 \sim q(\mathbf{x}_0)$. The joint distribution $p_\theta(\mathbf{x}_{0:T})$ is called the *reverse process*, and it is defined as a Markov chain with learned Gaussian transitions starting at $p(\mathbf{x}_T) = \mathcal{N}(\mathbf{x}_T; \mathbf{0}, \mathbf{I})$:

$$p_\theta(\mathbf{x}_{0:T}) \coloneqq p(\mathbf{x}_T) \prod_{t=1}^{T} p_\theta(\mathbf{x}_{t-1}|\mathbf{x}_t), \qquad p_\theta(\mathbf{x}_{t-1}|\mathbf{x}_t) \coloneqq \mathcal{N}(\mathbf{x}_{t-1}; \boldsymbol{\mu}_\theta(\mathbf{x}_t, t), \boldsymbol{\Sigma}_\theta(\mathbf{x}_t, t)) \quad (1)$$

What distinguishes diffusion models from other types of latent variable models is that the approximate posterior $q(\mathbf{x}_{1:T}|\mathbf{x}_0)$, called the *forward process* or *diffusion process*, is fixed to a Markov chain that gradually adds Gaussian noise to the data according to a variance schedule $\beta_1, \ldots, \beta_T$:

$$q(\mathbf{x}_{1:T}|\mathbf{x}_0) \coloneqq \prod_{t=1}^{T} q(\mathbf{x}_t|\mathbf{x}_{t-1}), \qquad q(\mathbf{x}_t|\mathbf{x}_{t-1}) \coloneqq \mathcal{N}(\mathbf{x}_t; \sqrt{1-\beta_t}\mathbf{x}_{t-1}, \beta_t \mathbf{I}) \quad (2)$$

Training is performed by optimizing the usual variational bound on negative log likelihood:

$$\mathbb{E}\left[-\log p_\theta(\mathbf{x}_0)\right] \leq \mathbb{E}_q\left[-\log \frac{p_\theta(\mathbf{x}_{0:T})}{q(\mathbf{x}_{1:T}|\mathbf{x}_0)}\right] = \mathbb{E}_q\left[-\log p(\mathbf{x}_T) - \sum_{t\geq 1}\log \frac{p_\theta(\mathbf{x}_{t-1}|\mathbf{x}_t)}{q(\mathbf{x}_t|\mathbf{x}_{t-1})}\right] =: L \quad (3)$$

The forward process variances $\beta_t$ can be learned by reparameterization [33] or held constant as hyperparameters, and expressiveness of the reverse process is ensured in part by the choice of Gaussian conditionals in $p_\theta(\mathbf{x}_{t-1}|\mathbf{x}_t)$, because both processes have the same functional form when $\beta_t$ are small [53]. A notable property of the forward process is that it admits sampling $\mathbf{x}_t$ at an arbitrary timestep $t$ in closed form: using the notation $\alpha_t \coloneqq 1 - \beta_t$ and $\bar{\alpha}_t \coloneqq \prod_{s=1}^{t} \alpha_s$, we have

$$q(\mathbf{x}_t|\mathbf{x}_0) = \mathcal{N}(\mathbf{x}_t; \sqrt{\bar{\alpha}_t}\mathbf{x}_0, (1-\bar{\alpha}_t)\mathbf{I}) \quad (4)$$

Efficient training is therefore possible by optimizing random terms of $L$ with stochastic gradient descent. Further improvements come from variance reduction by rewriting $L$ (3) as:

$$\mathbb{E}_q\left[\underbrace{D_{\mathrm{KL}}(q(\mathbf{x}_T|\mathbf{x}_0) \parallel p(\mathbf{x}_T))}_{L_T} + \sum_{t>1}\underbrace{D_{\mathrm{KL}}(q(\mathbf{x}_{t-1}|\mathbf{x}_t,\mathbf{x}_0) \parallel p_\theta(\mathbf{x}_{t-1}|\mathbf{x}_t))}_{L_{t-1}} \underbrace{- \log p_\theta(\mathbf{x}_0|\mathbf{x}_1)}_{L_0}\right] \quad (5)$$

(See Appendix A for details. The labels on the terms are used in Section 3.) Equation (5) uses KL divergence to directly compare $p_\theta(\mathbf{x}_{t-1}|\mathbf{x}_t)$ against forward process posteriors, which are tractable when conditioned on $\mathbf{x}_0$:

$$q(\mathbf{x}_{t-1}|\mathbf{x}_t,\mathbf{x}_0) = \mathcal{N}(\mathbf{x}_{t-1};\tilde{\boldsymbol{\mu}}_t(\mathbf{x}_t,\mathbf{x}_0),\tilde{\beta}_t\mathbf{I}), \quad (6)$$

$$\text{where} \quad \tilde{\boldsymbol{\mu}}_t(\mathbf{x}_t,\mathbf{x}_0) := \frac{\sqrt{\bar{\alpha}_{t-1}}\beta_t}{1-\bar{\alpha}_t}\mathbf{x}_0 + \frac{\sqrt{\alpha_t}(1-\bar{\alpha}_{t-1})}{1-\bar{\alpha}_t}\mathbf{x}_t \quad \text{and} \quad \tilde{\beta}_t := \frac{1-\bar{\alpha}_{t-1}}{1-\bar{\alpha}_t}\beta_t \quad (7)$$

Consequently, all KL divergences in Eq. (5) are comparisons between Gaussians, so they can be calculated in a Rao-Blackwellized fashion with closed form expressions instead of high variance Monte Carlo estimates.

## 3   Diffusion models and denoising autoencoders

Diffusion models might appear to be a restricted class of latent variable models, but they allow a large number of degrees of freedom in implementation. One must choose the variances $\beta_t$ of the forward process and the model architecture and Gaussian distribution parameterization of the reverse process. To guide our choices, we establish a new explicit connection between diffusion models and denoising score matching (Section 3.2) that leads to a simplified, weighted variational bound objective for diffusion models (Section 3.4). Ultimately, our model design is justified by simplicity and empirical results (Section 4). Our discussion is categorized by the terms of Eq. (5).

### 3.1   Forward process and $L_T$

We ignore the fact that the forward process variances $\beta_t$ are learnable by reparameterization and instead fix them to constants (see Section 4 for details). Thus, in our implementation, the approximate posterior $q$ has no learnable parameters, so $L_T$ is a constant during training and can be ignored.

### 3.2   Reverse process and $L_{1:T-1}$

Now we discuss our choices in $p_\theta(\mathbf{x}_{t-1}|\mathbf{x}_t) = \mathcal{N}(\mathbf{x}_{t-1};\boldsymbol{\mu}_\theta(\mathbf{x}_t,t),\boldsymbol{\Sigma}_\theta(\mathbf{x}_t,t))$ for $1 < t \le T$. First, we set $\boldsymbol{\Sigma}_\theta(\mathbf{x}_t,t) = \sigma_t^2\mathbf{I}$ to untrained time dependent constants. Experimentally, both $\sigma_t^2 = \beta_t$ and $\sigma_t^2 = \tilde{\beta}_t = \frac{1-\bar{\alpha}_{t-1}}{1-\bar{\alpha}_t}\beta_t$ had similar results. The first choice is optimal for $\mathbf{x}_0 \sim \mathcal{N}(\mathbf{0},\mathbf{I})$, and the second is optimal for $\mathbf{x}_0$ deterministically set to one point. These are the two extreme choices corresponding to upper and lower bounds on reverse process entropy for data with coordinatewise unit variance [53].

Second, to represent the mean $\boldsymbol{\mu}_\theta(\mathbf{x}_t,t)$, we propose a specific parameterization motivated by the following analysis of $L_t$. With $p_\theta(\mathbf{x}_{t-1}|\mathbf{x}_t) = \mathcal{N}(\mathbf{x}_{t-1};\boldsymbol{\mu}_\theta(\mathbf{x}_t,t),\sigma_t^2\mathbf{I})$, we can write:

$$L_{t-1} = \mathbb{E}_q\left[\frac{1}{2\sigma_t^2}\|\tilde{\boldsymbol{\mu}}_t(\mathbf{x}_t,\mathbf{x}_0) - \boldsymbol{\mu}_\theta(\mathbf{x}_t,t)\|^2\right] + C \quad (8)$$

where $C$ is a constant that does not depend on $\theta$. So, we see that the most straightforward parameterization of $\boldsymbol{\mu}_\theta$ is a model that predicts $\tilde{\boldsymbol{\mu}}_t$, the forward process posterior mean. However, we can expand Eq. (8) further by reparameterizing Eq. (4) as $\mathbf{x}_t(\mathbf{x}_0,\boldsymbol{\epsilon}) = \sqrt{\bar{\alpha}_t}\mathbf{x}_0 + \sqrt{1-\bar{\alpha}_t}\boldsymbol{\epsilon}$ for $\boldsymbol{\epsilon} \sim \mathcal{N}(\mathbf{0},\mathbf{I})$ and applying the forward process posterior formula (7):

$$L_{t-1} - C = \mathbb{E}_{\mathbf{x}_0,\boldsymbol{\epsilon}}\left[\frac{1}{2\sigma_t^2}\left\|\tilde{\boldsymbol{\mu}}_t\left(\mathbf{x}_t(\mathbf{x}_0,\boldsymbol{\epsilon}),\frac{1}{\sqrt{\bar{\alpha}_t}}(\mathbf{x}_t(\mathbf{x}_0,\boldsymbol{\epsilon}) - \sqrt{1-\bar{\alpha}_t}\boldsymbol{\epsilon})\right) - \boldsymbol{\mu}_\theta(\mathbf{x}_t(\mathbf{x}_0,\boldsymbol{\epsilon}),t)\right\|^2\right]$$
$$(9)$$

$$= \mathbb{E}_{\mathbf{x}_0,\boldsymbol{\epsilon}}\left[\frac{1}{2\sigma_t^2}\left\|\frac{1}{\sqrt{\alpha_t}}\left(\mathbf{x}_t(\mathbf{x}_0,\boldsymbol{\epsilon}) - \frac{\beta_t}{\sqrt{1-\bar{\alpha}_t}}\boldsymbol{\epsilon}\right) - \boldsymbol{\mu}_\theta(\mathbf{x}_t(\mathbf{x}_0,\boldsymbol{\epsilon}),t)\right\|^2\right] \quad (10)$$

| **Algorithm 1** Training | **Algorithm 2** Sampling |
|---|---|
| 1: **repeat** | 1: $\mathbf{x}_T \sim \mathcal{N}(\mathbf{0}, \mathbf{I})$ |
| 2:   $\mathbf{x}_0 \sim q(\mathbf{x}_0)$ | 2: **for** $t = T, \ldots, 1$ **do** |
| 3:   $t \sim \mathrm{Uniform}(\{1, \ldots, T\})$ | 3:   $\mathbf{z} \sim \mathcal{N}(\mathbf{0}, \mathbf{I})$ if $t > 1$, else $\mathbf{z} = \mathbf{0}$ |
| 4:   $\boldsymbol{\epsilon} \sim \mathcal{N}(\mathbf{0}, \mathbf{I})$ | 4:   $\mathbf{x}_{t-1} = \frac{1}{\sqrt{\alpha_t}} \left( \mathbf{x}_t - \frac{1-\alpha_t}{\sqrt{1-\bar{\alpha}_t}} \boldsymbol{\epsilon}_\theta(\mathbf{x}_t, t) \right) + \sigma_t \mathbf{z}$ |
| 5:   Take gradient descent step on | 5: **end for** |
| $\qquad \nabla_\theta \left\| \boldsymbol{\epsilon} - \boldsymbol{\epsilon}_\theta(\sqrt{\bar{\alpha}_t}\mathbf{x}_0 + \sqrt{1-\bar{\alpha}_t}\boldsymbol{\epsilon}, t) \right\|^2$ | 6: **return** $\mathbf{x}_0$ |
| 6: **until** converged | |

Equation (10) reveals that $\boldsymbol{\mu}_\theta$ must predict $\frac{1}{\sqrt{\alpha_t}} \left( \mathbf{x}_t - \frac{\beta_t}{\sqrt{1-\bar{\alpha}_t}} \boldsymbol{\epsilon} \right)$ given $\mathbf{x}_t$. Since $\mathbf{x}_t$ is available as input to the model, we may choose the parameterization

$$\boldsymbol{\mu}_\theta(\mathbf{x}_t, t) = \tilde{\boldsymbol{\mu}}_t \left( \mathbf{x}_t, \frac{1}{\sqrt{\bar{\alpha}_t}}(\mathbf{x}_t - \sqrt{1-\bar{\alpha}_t}\boldsymbol{\epsilon}_\theta(\mathbf{x}_t)) \right) = \frac{1}{\sqrt{\alpha_t}} \left( \mathbf{x}_t - \frac{\beta_t}{\sqrt{1-\bar{\alpha}_t}} \boldsymbol{\epsilon}_\theta(\mathbf{x}_t, t) \right) \qquad (11)$$

where $\boldsymbol{\epsilon}_\theta$ is a function approximator intended to predict $\boldsymbol{\epsilon}$ from $\mathbf{x}_t$. To sample $\mathbf{x}_{t-1} \sim p_\theta(\mathbf{x}_{t-1}|\mathbf{x}_t)$ is to compute $\mathbf{x}_{t-1} = \frac{1}{\sqrt{\alpha_t}} \left( \mathbf{x}_t - \frac{\beta_t}{\sqrt{1-\bar{\alpha}_t}} \boldsymbol{\epsilon}_\theta(\mathbf{x}_t, t) \right) + \sigma_t \mathbf{z}$, where $\mathbf{z} \sim \mathcal{N}(\mathbf{0}, \mathbf{I})$. The complete sampling procedure, Algorithm 2, resembles Langevin dynamics with $\boldsymbol{\epsilon}_\theta$ as a learned gradient of the data density. Furthermore, with the parameterization (11), Eq. (10) simplifies to:

$$\mathbb{E}_{\mathbf{x}_0, \boldsymbol{\epsilon}} \left[ \frac{\beta_t^2}{2\sigma_t^2 \alpha_t (1-\bar{\alpha}_t)} \left\| \boldsymbol{\epsilon} - \boldsymbol{\epsilon}_\theta(\sqrt{\bar{\alpha}_t}\mathbf{x}_0 + \sqrt{1-\bar{\alpha}_t}\boldsymbol{\epsilon}, t) \right\|^2 \right] \qquad (12)$$

which resembles denoising score matching over multiple noise scales indexed by $t$ [55]. As Eq. (12) is equal to (one term of) the variational bound for the Langevin-like reverse process (11), we see that optimizing an objective resembling denoising score matching is equivalent to using variational inference to fit the finite-time marginal of a sampling chain resembling Langevin dynamics.

To summarize, we can train the reverse process mean function approximator $\boldsymbol{\mu}_\theta$ to predict $\tilde{\boldsymbol{\mu}}_t$, or by modifying its parameterization, we can train it to predict $\boldsymbol{\epsilon}$. (There is also the possibility of predicting $\mathbf{x}_0$, but we found this to lead to worse sample quality early in our experiments.) We have shown that the $\boldsymbol{\epsilon}$-prediction parameterization both resembles Langevin dynamics and simplifies the diffusion model's variational bound to an objective that resembles denoising score matching. Nonetheless, it is just another parameterization of $p_\theta(\mathbf{x}_{t-1}|\mathbf{x}_t)$, so we verify its effectiveness in Section 4 in an ablation where we compare predicting $\boldsymbol{\epsilon}$ against predicting $\tilde{\boldsymbol{\mu}}_t$.

### 3.3   Data scaling, reverse process decoder, and $L_0$

We assume that image data consists of integers in $\{0, 1, \ldots, 255\}$ scaled linearly to $[-1, 1]$. This ensures that the neural network reverse process operates on consistently scaled inputs starting from the standard normal prior $p(\mathbf{x}_T)$. To obtain discrete log likelihoods, we set the last term of the reverse process to an independent discrete decoder derived from the Gaussian $\mathcal{N}(\mathbf{x}_0; \boldsymbol{\mu}_\theta(\mathbf{x}_1, 1), \sigma_1^2 \mathbf{I})$:

$$p_\theta(\mathbf{x}_0|\mathbf{x}_1) = \prod_{i=1}^{D} \int_{\delta_-(x_0^i)}^{\delta_+(x_0^i)} \mathcal{N}(x; \mu_\theta^i(\mathbf{x}_1, 1), \sigma_1^2)\, dx$$
$$\delta_+(x) = \begin{cases} \infty & \text{if } x = 1 \\ x + \frac{1}{255} & \text{if } x < 1 \end{cases} \qquad \delta_-(x) = \begin{cases} -\infty & \text{if } x = -1 \\ x - \frac{1}{255} & \text{if } x > -1 \end{cases} \qquad (13)$$

where $D$ is the data dimensionality and the $i$ superscript indicates extraction of one coordinate. (It would be straightforward to instead incorporate a more powerful decoder like a conditional autoregressive model, but we leave that to future work.) Similar to the discretized continuous distributions used in VAE decoders and autoregressive models [34, 52], our choice here ensures that the variational bound is a lossless codelength of discrete data, without need of adding noise to the data or incorporating the Jacobian of the scaling operation into the log likelihood. At the end of sampling, we display $\boldsymbol{\mu}_\theta(\mathbf{x}_1, 1)$ noiselessly.

### 3.4   Simplified training objective

With the reverse process and decoder defined above, the variational bound, consisting of terms derived from Eqs. (12) and (13), is clearly differentiable with respect to $\theta$ and is ready to be employed for

Table 1: CIFAR10 results. NLL measured in bits/dim.

| Model | IS | FID | NLL Test (Train) |
|---|---|---|---|
| **Conditional** | | | |
| EBM [11] | 8.30 | 37.9 | |
| JEM [17] | 8.76 | 38.4 | |
| BigGAN [3] | 9.22 | 14.73 | |
| StyleGAN2 + ADA (v1) [29] | **10.06** | **2.67** | |
| **Unconditional** | | | |
| Diffusion (original) [53] | | | $\leq 5.40$ |
| Gated PixelCNN [59] | 4.60 | 65.93 | 3.03 (2.90) |
| Sparse Transformer [7] | | | **2.80** |
| PixelIQN [43] | 5.29 | 49.46 | |
| EBM [11] | 6.78 | 38.2 | |
| NCSNv2 [56] | | 31.75 | |
| NCSN [55] | 8.87±0.12 | 25.32 | |
| SNGAN [39] | 8.22±0.05 | 21.7 | |
| SNGAN-DDLS [4] | 9.09±0.10 | 15.42 | |
| StyleGAN2 + ADA (v1) [29] | **9.74** ± 0.05 | 3.26 | |
| Ours ($L$, fixed isotropic $\mathbf{\Sigma}$) | 7.67±0.13 | 13.51 | $\leq 3.70$ (3.69) |
| **Ours** ($L_{\text{simple}}$) | 9.46±0.11 | **3.17** | $\leq 3.75$ (3.72) |

Table 2: Unconditional CIFAR10 reverse process parameterization and training objective ablation. Blank entries were unstable to train and generated poor samples with out-of-range scores.

| Objective | IS | FID |
|---|---|---|
| $\tilde{\boldsymbol{\mu}}$ **prediction (baseline)** | | |
| $L$, learned diagonal $\mathbf{\Sigma}$ | 7.28±0.10 | 23.69 |
| $L$, fixed isotropic $\mathbf{\Sigma}$ | 8.06±0.09 | 13.22 |
| $\|\tilde{\boldsymbol{\mu}} - \tilde{\boldsymbol{\mu}}_\theta\|^2$ | – | – |
| $\boldsymbol{\epsilon}$ **prediction (ours)** | | |
| $L$, learned diagonal $\mathbf{\Sigma}$ | – | – |
| $L$, fixed isotropic $\mathbf{\Sigma}$ | 7.67±0.13 | 13.51 |
| $\|\tilde{\boldsymbol{\epsilon}} - \boldsymbol{\epsilon}_\theta\|^2$ ($L_{\text{simple}}$) | **9.46±0.11** | **3.17** |

training. However, we found it beneficial to sample quality (and simpler to implement) to train on the following variant of the variational bound:

$$L_{\text{simple}}(\theta) := \mathbb{E}_{t,\mathbf{x}_0,\boldsymbol{\epsilon}}\left[\left\|\boldsymbol{\epsilon} - \boldsymbol{\epsilon}_\theta(\sqrt{\bar{\alpha}_t}\mathbf{x}_0 + \sqrt{1-\bar{\alpha}_t}\boldsymbol{\epsilon}, t)\right\|^2\right] \tag{14}$$

where $t$ is uniform between 1 and $T$. The $t = 1$ case corresponds to $L_0$ with the integral in the discrete decoder definition (13) approximated by the Gaussian probability density function times the bin width, ignoring $\sigma_1^2$ and edge effects. The $t > 1$ cases correspond to an unweighted version of Eq. (12), analogous to the loss weighting used by the NCSN denoising score matching model [55]. ($L_T$ does not appear because the forward process variances $\beta_t$ are fixed.) Algorithm 1 displays the complete training procedure with this simplified objective.

Since our simplified objective (14) discards the weighting in Eq. (12), it is a weighted variational bound that emphasizes different aspects of reconstruction compared to the standard variational bound [18, 22]. In particular, our diffusion process setup in Section 4 causes the simplified objective to down-weight loss terms corresponding to small $t$. These terms train the network to denoise data with very small amounts of noise, so it is beneficial to down-weight them so that the network can focus on more difficult denoising tasks at larger $t$ terms. We will see in our experiments that this reweighting leads to better sample quality.

## 4 Experiments

We set $T = 1000$ for all experiments so that the number of neural network evaluations needed during sampling matches previous work [53, 55]. We set the forward process variances to constants increasing linearly from $\beta_1 = 10^{-4}$ to $\beta_T = 0.02$. These constants were chosen to be small relative to data scaled to $[-1, 1]$, ensuring that reverse and forward processes have approximately the same functional form while keeping the signal-to-noise ratio at $\mathbf{x}_T$ as small as possible ($L_T = D_{\text{KL}}(q(\mathbf{x}_T|\mathbf{x}_0) \| \mathcal{N}(\mathbf{0}, \mathbf{I})) \approx 10^{-5}$ bits per dimension in our experiments).

To represent the reverse process, we use a U-Net backbone similar to an unmasked PixelCNN++ [52, 48] with group normalization throughout [66]. Parameters are shared across time, which is specified to the network using the Transformer sinusoidal position embedding [60]. We use self-attention at the $16 \times 16$ feature map resolution [63, 60]. Details are in Appendix B.

### 4.1 Sample quality

Table 1 shows Inception scores, FID scores, and negative log likelihoods (lossless codelengths) on CIFAR10. With our FID score of 3.17, our unconditional model achieves better sample quality than most models in the literature, including class conditional models. Our FID score is computed with respect to the training set, as is standard practice; when we compute it with respect to the test set, the score is 5.24, which is still better than many of the training set FID scores in the literature.

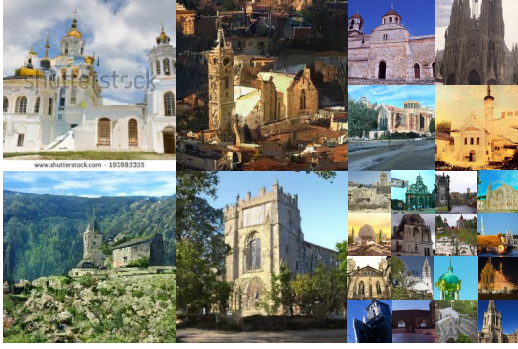 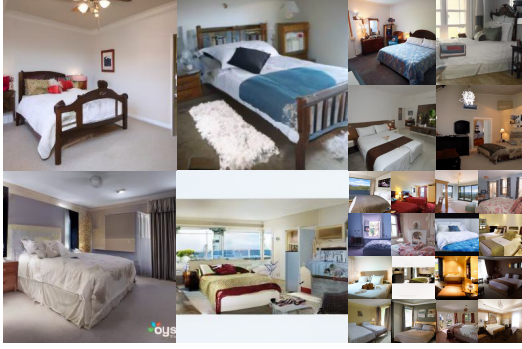

Figure 3: LSUN Church samples. FID=7.89      Figure 4: LSUN Bedroom samples. FID=4.90

| **Algorithm 3** Sending $\mathbf{x}_0$ | **Algorithm 4** Receiving |
|---|---|
| 1: Send $\mathbf{x}_T \sim q(\mathbf{x}_T \mid \mathbf{x}_0)$ using $p(\mathbf{x}_T)$ | 1: Receive $\mathbf{x}_T$ using $p(\mathbf{x}_T)$ |
| 2: **for** $t = T-1, \ldots, 2, 1$ **do** | 2: **for** $t = T-1, \ldots, 1, 0$ **do** |
| 3:    Send $\mathbf{x}_t \sim q(\mathbf{x}_t \mid \mathbf{x}_{t+1}, \mathbf{x}_0)$ using $p_\theta(\mathbf{x}_t \mid \mathbf{x}_{t+1})$ | 3:    Receive $\mathbf{x}_t$ using $p_\theta(\mathbf{x}_t \mid \mathbf{x}_{t+1})$ |
| 4: **end for** | 4: **end for** |
| 5: Send $\mathbf{x}_0$ using $p_\theta(\mathbf{x}_0 \mid \mathbf{x}_1)$ | 5: **return** $\mathbf{x}_0$ |

We find that training our models on the true variational bound yields better codelengths than training on the simplified objective, as expected, but the latter yields the best sample quality. See Fig. 1 for CIFAR10 and CelebA-HQ $256 \times 256$ samples, Fig. 3 and Fig. 4 for LSUN $256 \times 256$ samples [71], and Appendix D for more.

### 4.2 Reverse process parameterization and training objective ablation

In Table 2, we show the sample quality effects of reverse process parameterizations and training objectives (Section 3.2). We find that the baseline option of predicting $\tilde{\boldsymbol{\mu}}$ works well only when trained on the true variational bound instead of unweighted mean squared error, a simplified objective akin to Eq. (14). We also see that learning reverse process variances (by incorporating a parameterized diagonal $\boldsymbol{\Sigma}_\theta(\mathbf{x}_t)$ into the variational bound) leads to unstable training and poorer sample quality compared to fixed variances. Predicting $\boldsymbol{\epsilon}$, as we proposed, performs approximately as well as predicting $\tilde{\boldsymbol{\mu}}$ when trained on the variational bound with fixed variances, but much better when trained with our simplified objective.

### 4.3 Progressive coding

Table 1 also shows the codelengths of our CIFAR10 models. The gap between train and test is at most 0.03 bits per dimension, which is comparable to the gaps reported with other likelihood-based models and indicates that our diffusion model is not overfitting (see Appendix D for nearest neighbor visualizations). Still, while our lossless codelengths are better than the large estimates reported for energy based models and score matching using annealed importance sampling [11], they are not competitive with other types of likelihood-based generative models [7].

Since our samples are nonetheless of high quality, we conclude that diffusion models have an inductive bias that makes them excellent lossy compressors. Treating the variational bound terms $L_1 + \cdots + L_T$ as rate and $L_0$ as distortion, our CIFAR10 model with the highest quality samples has a rate of **1.78** bits/dim and a distortion of **1.97** bits/dim, which amounts to a root mean squared error of 0.95 on a scale from 0 to 255. More than half of the lossless codelength describes imperceptible distortions.

**Progressive lossy compression**    We can probe further into the rate-distortion behavior of our model by introducing a progressive lossy code that mirrors the form of Eq. (5): see Algorithms 3 and 4, which assume access to a procedure, such as minimal random coding [19, 20], that can transmit a sample $\mathbf{x} \sim q(\mathbf{x})$ using approximately $D_{\mathrm{KL}}(q(\mathbf{x}) \parallel p(\mathbf{x}))$ bits on average for any distributions $p$ and $q$, for which only $p$ is available to the receiver beforehand. When applied to $\mathbf{x}_0 \sim q(\mathbf{x}_0)$, Algorithms 3 and 4 transmit $\mathbf{x}_T, \ldots, \mathbf{x}_0$ in sequence using a total expected codelength equal to Eq. (5). The receiver,

at any time $t$, has the partial information $\mathbf{x}_t$ fully available and can progressively estimate:

$$\mathbf{x}_0 \approx \hat{\mathbf{x}}_0 = \left(\mathbf{x}_t - \sqrt{1 - \bar{\alpha}_t}\boldsymbol{\epsilon}_\theta(\mathbf{x}_t)\right)/\sqrt{\bar{\alpha}_t} \qquad (15)$$

due to Eq. (4). (A stochastic reconstruction $\mathbf{x}_0 \sim p_\theta(\mathbf{x}_0|\mathbf{x}_t)$ is also valid, but we do not consider it here because it makes distortion more difficult to evaluate.) Figure 5 shows the resulting rate-distortion plot on the CIFAR10 test set. At each time $t$, the distortion is calculated as the root mean squared error $\sqrt{\|\mathbf{x}_0 - \hat{\mathbf{x}}_0\|^2/D}$, and the rate is calculated as the cumulative number of bits received so far at time $t$. The distortion decreases steeply in the low-rate region of the rate-distortion plot, indicating that the majority of the bits are indeed allocated to imperceptible distortions.

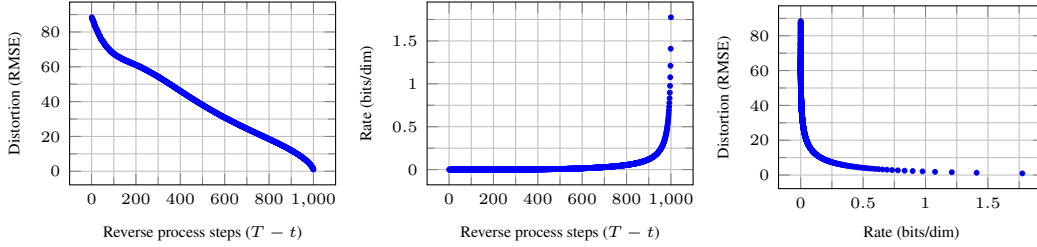

Figure 5: Unconditional CIFAR10 test set rate-distortion vs. time. Distortion is measured in root mean squared error on a $[0, 255]$ scale. See Table 4 for details.

**Progressive generation**    We also run a progressive unconditional generation process given by progressive decompression from random bits. In other words, we predict the result of the reverse process, $\hat{\mathbf{x}}_0$, while sampling from the reverse process using Algorithm 2. Figures 6 and 10 show the resulting sample quality of $\hat{\mathbf{x}}_0$ over the course of the reverse process. Large scale image features appear first and details appear last. Figure 7 shows stochastic predictions $\mathbf{x}_0 \sim p_\theta(\mathbf{x}_0|\mathbf{x}_t)$ with $\mathbf{x}_t$ frozen for various $t$. When $t$ is small, all but fine details are preserved, and when $t$ is large, only large scale features are preserved. Perhaps these are hints of conceptual compression [18].

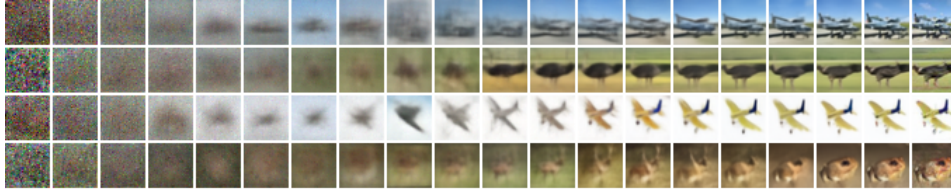

Figure 6: Unconditional CIFAR10 progressive generation ($\hat{\mathbf{x}}_0$ over time, from left to right). Extended samples and sample quality metrics over time in the appendix (Figs. 10 and 14).

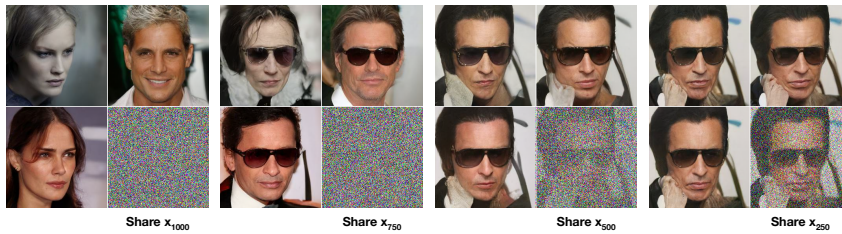

Figure 7: When conditioned on the same latent, CelebA-HQ $256 \times 256$ samples share high-level attributes. Bottom-right quadrants are $\mathbf{x}_t$, and other quadrants are samples from $p_\theta(\mathbf{x}_0|\mathbf{x}_t)$.

**Connection to autoregressive decoding**    Note that the variational bound (5) can be rewritten as:

$$L = D_{\mathrm{KL}}(q(\mathbf{x}_T) \,\|\, p(\mathbf{x}_T)) + \mathbb{E}_q\left[\sum_{t \geq 1} D_{\mathrm{KL}}(q(\mathbf{x}_{t-1}|\mathbf{x}_t) \,\|\, p_\theta(\mathbf{x}_{t-1}|\mathbf{x}_t))\right] + H(\mathbf{x}_0) \qquad (16)$$

(See Appendix A for a derivation.) Now consider setting the diffusion process length $T$ to the dimensionality of the data, defining the forward process so that $q(\mathbf{x}_t|\mathbf{x}_0)$ places all probability mass on $\mathbf{x}_0$ with the first $t$ coordinates masked out (i.e. $q(\mathbf{x}_t|\mathbf{x}_{t-1})$ masks out the $t^{\text{th}}$ coordinate), setting

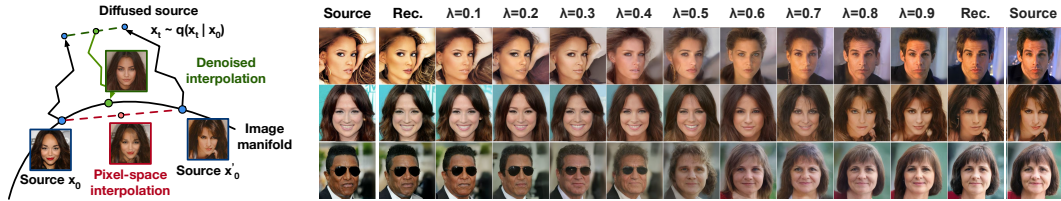

Figure 8: Interpolations of CelebA-HQ 256x256 images with 500 timesteps of diffusion.

$p(\mathbf{x}_T)$ to place all mass on a blank image, and, for the sake of argument, taking $p_\theta(\mathbf{x}_{t-1}|\mathbf{x}_t)$ to be a fully expressive conditional distribution. With these choices, $D_{\mathrm{KL}}(q(\mathbf{x}_T) \,\|\, p(\mathbf{x}_T)) = 0$, and minimizing $D_{\mathrm{KL}}(q(\mathbf{x}_{t-1}|\mathbf{x}_t) \,\|\, p_\theta(\mathbf{x}_{t-1}|\mathbf{x}_t))$ trains $p_\theta$ to copy coordinates $t + 1, \ldots, T$ unchanged and to predict the $t^{\mathrm{th}}$ coordinate given $t + 1, \ldots, T$. Thus, training $p_\theta$ with this particular diffusion is training an autoregressive model.

We can therefore interpret the Gaussian diffusion model (2) as a kind of autoregressive model with a generalized bit ordering that cannot be expressed by reordering data coordinates. Prior work has shown that such reorderings introduce inductive biases that have an impact on sample quality [38], so we speculate that the Gaussian diffusion serves a similar purpose, perhaps to greater effect since Gaussian noise might be more natural to add to images compared to masking noise. Moreover, the Gaussian diffusion length is not restricted to equal the data dimension; for instance, we use $T = 1000$, which is less than the dimension of the $32 \times 32 \times 3$ or $256 \times 256 \times 3$ images in our experiments. Gaussian diffusions can be made shorter for fast sampling or longer for model expressiveness.

## 4.4 Interpolation

We can interpolate source images $\mathbf{x}_0, \mathbf{x}_0' \sim q(\mathbf{x}_0)$ in latent space using $q$ as a stochastic encoder, $\mathbf{x}_t, \mathbf{x}_t' \sim q(\mathbf{x}_t|\mathbf{x}_0)$, then decoding the linearly interpolated latent $\bar{\mathbf{x}}_t = (1 - \lambda)\mathbf{x}_0 + \lambda\mathbf{x}_0'$ into image space by the reverse process, $\bar{\mathbf{x}}_0 \sim p(\mathbf{x}_0|\bar{\mathbf{x}}_t)$. In effect, we use the reverse process to remove artifacts from linearly interpolating corrupted versions of the source images, as depicted in Fig. 8 (left). We fixed the noise for different values of $\lambda$ so $\mathbf{x}_t$ and $\mathbf{x}_t'$ remain the same. Fig. 8 (right) shows interpolations and reconstructions of original CelebA-HQ $256 \times 256$ images ($t = 500$). The reverse process produces high-quality reconstructions, and plausible interpolations that smoothly vary attributes such as pose, skin tone, hairstyle, expression and background, but not eyewear. Larger $t$ results in coarser and more varied interpolations, with novel samples at $t = 1000$ (Appendix Fig. 9).

## 5 Related Work

While diffusion models might resemble flows [9, 46, 10, 32, 5, 16, 23] and VAEs [33, 47, 37], diffusion models are designed so that $q$ has no parameters and the top-level latent $\mathbf{x}_T$ has nearly zero mutual information with the data $\mathbf{x}_0$. Our $\boldsymbol{\epsilon}$-prediction reverse process parameterization establishes a connection between diffusion models and denoising score matching over multiple noise levels with annealed Langevin dynamics for sampling [55, 56]. Diffusion models, however, admit straightforward log likelihood evaluation, and the training procedure explicitly trains the Langevin dynamics sampler using variational inference (see Appendix C for details). The connection also has the reverse implication that a certain weighted form of denoising score matching is the same as variational inference to train a Langevin-like sampler. Other methods for learning transition operators of Markov chains include infusion training [2], variational walkback [15], generative stochastic networks [1], and others [50, 54, 36, 42, 35, 65].

By the known connection between score matching and energy-based modeling, our work could have implications for other recent work on energy-based models [67–69, 12, 70, 13, 11, 41, 17, 8]. Our rate-distortion curves are computed over time in one evaluation of the variational bound, reminiscent of how rate-distortion curves can be computed over distortion penalties in one run of annealed importance sampling [24]. Our progressive decoding argument can be seen in convolutional DRAW and related models [18, 40] and may also lead to more general designs for subscale orderings or sampling strategies for autoregressive models [38, 64].

# 6  Conclusion

We have presented high quality image samples using diffusion models, and we have found connections among diffusion models and variational inference for training Markov chains, denoising score matching and annealed Langevin dynamics (and energy-based models by extension), autoregressive models, and progressive lossy compression. Since diffusion models seem to have excellent inductive biases for image data, we look forward to investigating their utility in other data modalities and as components in other types of generative models and machine learning systems.

## Broader Impact

Our work on diffusion models takes on a similar scope as existing work on other types of deep generative models, such as efforts to improve the sample quality of GANs, flows, autoregressive models, and so forth. Our paper represents progress in making diffusion models a generally useful tool in this family of techniques, so it may serve to amplify any impacts that generative models have had (and will have) on the broader world.

Unfortunately, there are numerous well-known malicious uses of generative models. Sample generation techniques can be employed to produce fake images and videos of high profile figures for political purposes. While fake images were manually created long before software tools were available, generative models such as ours make the process easier. Fortunately, CNN-generated images currently have subtle flaws that allow detection [62], but improvements in generative models may make this more difficult. Generative models also reflect the biases in the datasets on which they are trained. As many large datasets are collected from the internet by automated systems, it can be difficult to remove these biases, especially when the images are unlabeled. If samples from generative models trained on these datasets proliferate throughout the internet, then these biases will only be reinforced further.

On the other hand, diffusion models may be useful for data compression, which, as data becomes higher resolution and as global internet traffic increases, might be crucial to ensure accessibility of the internet to wide audiences. Our work might contribute to representation learning on unlabeled raw data for a large range of downstream tasks, from image classification to reinforcement learning, and diffusion models might also become viable for creative uses in art, photography, and music.

## Acknowledgments and Disclosure of Funding

This work was supported by ONR PECASE and the NSF Graduate Research Fellowship under grant number DGE-1752814. Google's TensorFlow Research Cloud (TFRC) provided Cloud TPUs.

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
