[Supplementary Material]

# Extra information

**LSUN**  FID scores for LSUN datasets are included in Table 3. Scores marked with $^*$ are reported by StyleGAN2 as baselines, and other scores are reported by their respective authors.

Table 3: FID scores for LSUN $256 \times 256$ datasets

| Model | LSUN Bedroom | LSUN Church | LSUN Cat |
|---|---|---|---|
| ProgressiveGAN [27] | 8.34 | 6.42 | 37.52 |
| StyleGAN [28] | **2.65** | $4.21^*$ | $8.53^*$ |
| StyleGAN2 [30] | - | **3.86** | **6.93** |
| Ours ($L_{\text{simple}}$) | 6.36 | 7.89 | 19.75 |
| Ours ($L_{\text{simple}}$, large) | 4.90 | - | - |

**Progressive compression**  Our lossy compression argument in Section 4.3 is only a proof of concept, because Algorithms 3 and 4 depend on a procedure such as minimal random coding [20], which is not tractable for high dimensional data. These algorithms serve as a compression interpretation of the variational bound (5) of Sohl-Dickstein et al. [53], not yet as a practical compression system.

Table 4: Unconditional CIFAR10 test set rate-distortion values (accompanies Fig. 5)

| Reverse process time $(T - t + 1)$ | Rate (bits/dim) | Distortion (RMSE $[0, 255]$) |
|---|---|---|
| 1000 | 1.77581 | 0.95136 |
| 900 | 0.11994 | 12.02277 |
| 800 | 0.05415 | 18.47482 |
| 700 | 0.02866 | 24.43656 |
| 600 | 0.01507 | 30.80948 |
| 500 | 0.00716 | 38.03236 |
| 400 | 0.00282 | 46.12765 |
| 300 | 0.00081 | 54.18826 |
| 200 | 0.00013 | 60.97170 |
| 100 | 0.00000 | 67.60125 |

# A  Extended derivations

Below is a derivation of Eq. (5), the reduced variance variational bound for diffusion models. This material is from Sohl-Dickstein et al. [53]; we include it here only for completeness.

$$L = \mathbb{E}_q \left[ - \log \frac{p_\theta(\mathbf{x}_{0:T})}{q(\mathbf{x}_{1:T}|\mathbf{x}_0)} \right] \tag{17}$$

$$= \mathbb{E}_q \left[ - \log p(\mathbf{x}_T) - \sum_{t \geq 1} \log \frac{p_\theta(\mathbf{x}_{t-1}|\mathbf{x}_t)}{q(\mathbf{x}_t|\mathbf{x}_{t-1})} \right] \tag{18}$$

$$= \mathbb{E}_q \left[ - \log p(\mathbf{x}_T) - \sum_{t > 1} \log \frac{p_\theta(\mathbf{x}_{t-1}|\mathbf{x}_t)}{q(\mathbf{x}_t|\mathbf{x}_{t-1})} - \log \frac{p_\theta(\mathbf{x}_0|\mathbf{x}_1)}{q(\mathbf{x}_1|\mathbf{x}_0)} \right] \tag{19}$$

$$= \mathbb{E}_q \left[ - \log p(\mathbf{x}_T) - \sum_{t > 1} \log \frac{p_\theta(\mathbf{x}_{t-1}|\mathbf{x}_t)}{q(\mathbf{x}_{t-1}|\mathbf{x}_t, \mathbf{x}_0)} \cdot \frac{q(\mathbf{x}_{t-1}|\mathbf{x}_0)}{q(\mathbf{x}_t|\mathbf{x}_0)} - \log \frac{p_\theta(\mathbf{x}_0|\mathbf{x}_1)}{q(\mathbf{x}_1|\mathbf{x}_0)} \right] \tag{20}$$

$$= \mathbb{E}_q \left[ - \log \frac{p(\mathbf{x}_T)}{q(\mathbf{x}_T|\mathbf{x}_0)} - \sum_{t > 1} \log \frac{p_\theta(\mathbf{x}_{t-1}|\mathbf{x}_t)}{q(\mathbf{x}_{t-1}|\mathbf{x}_t, \mathbf{x}_0)} - \log p_\theta(\mathbf{x}_0|\mathbf{x}_1) \right] \tag{21}$$

$$= \mathbb{E}_q \left[ D_{\mathrm{KL}}(q(\mathbf{x}_T|\mathbf{x}_0) \parallel p(\mathbf{x}_T)) + \sum_{t>1} D_{\mathrm{KL}}(q(\mathbf{x}_{t-1}|\mathbf{x}_t, \mathbf{x}_0) \parallel p_\theta(\mathbf{x}_{t-1}|\mathbf{x}_t)) - \log p_\theta(\mathbf{x}_0|\mathbf{x}_1) \right] \tag{22}$$

The following is an alternate version of $L$. It is not tractable to estimate, but it is useful for our discussion in Section 4.3.

$$L = \mathbb{E}_q \left[ -\log p(\mathbf{x}_T) - \sum_{t \geq 1} \log \frac{p_\theta(\mathbf{x}_{t-1}|\mathbf{x}_t)}{q(\mathbf{x}_t|\mathbf{x}_{t-1})} \right] \tag{23}$$

$$= \mathbb{E}_q \left[ -\log p(\mathbf{x}_T) - \sum_{t \geq 1} \log \frac{p_\theta(\mathbf{x}_{t-1}|\mathbf{x}_t)}{q(\mathbf{x}_{t-1}|\mathbf{x}_t)} \cdot \frac{q(\mathbf{x}_{t-1})}{q(\mathbf{x}_t)} \right] \tag{24}$$

$$= \mathbb{E}_q \left[ -\log \frac{p(\mathbf{x}_T)}{q(\mathbf{x}_T)} - \sum_{t \geq 1} \log \frac{p_\theta(\mathbf{x}_{t-1}|\mathbf{x}_t)}{q(\mathbf{x}_{t-1}|\mathbf{x}_t)} - \log q(\mathbf{x}_0) \right] \tag{25}$$

$$= D_{\mathrm{KL}}(q(\mathbf{x}_T) \parallel p(\mathbf{x}_T)) + \mathbb{E}_q \left[ \sum_{t \geq 1} D_{\mathrm{KL}}(q(\mathbf{x}_{t-1}|\mathbf{x}_t) \parallel p_\theta(\mathbf{x}_{t-1}|\mathbf{x}_t)) \right] + H(\mathbf{x}_0) \tag{26}$$

## B  Experimental details

Our neural network architecture follows the backbone of PixelCNN++ [52], which is a U-Net [48] based on a Wide ResNet [72]. We replaced weight normalization [49] with group normalization [66] to make the implementation simpler. Our $32 \times 32$ models use four feature map resolutions ($32 \times 32$ to $4 \times 4$), and our $256 \times 256$ models use six. All models have two convolutional residual blocks per resolution level and self-attention blocks at the $16 \times 16$ resolution between the convolutional blocks [6]. Diffusion time $t$ is specified by adding the Transformer sinusoidal position embedding [60] into each residual block. Our CIFAR10 model has 35.7 million parameters, and our LSUN and CelebA-HQ models have 114 million parameters. We also trained a larger variant of the LSUN Bedroom model with approximately 256 million parameters by increasing filter count.

We used TPU v3-8 (similar to 8 V100 GPUs) for all experiments. Our CIFAR model trains at 21 steps per second at batch size 128 (10.6 hours to train to completion at 800k steps), and sampling a batch of 256 images takes 17 seconds. Our CelebA-HQ/LSUN ($256^2$) models train at 2.2 steps per second at batch size 64, and sampling a batch of 128 images takes 300 seconds. We trained on CelebA-HQ for 0.5M steps, LSUN Bedroom for 2.4M steps, LSUN Cat for 1.8M steps, and LSUN Church for 1.2M steps. The larger LSUN Bedroom model was trained for 1.15M steps.

Apart from an initial choice of hyperparameters early on to make network size fit within memory constraints, we performed the majority of our hyperparameter search to optimize for CIFAR10 sample quality, then transferred the resulting settings over to the other datasets:

- We chose the $\beta_t$ schedule from a set of constant, linear, and quadratic schedules, all constrained so that $L_T \approx 0$. We set $T = 1000$ without a sweep, and we chose a linear schedule from $\beta_1 = 10^{-4}$ to $\beta_T = 0.02$.

- We set the dropout rate on CIFAR10 to 0.1 by sweeping over the values $\{0.1, 0.2, 0.3, 0.4\}$. Without dropout on CIFAR10, we obtained poorer samples reminiscent of the overfitting artifacts in an unregularized PixelCNN++ [52]. We set dropout rate on the other datasets to zero without sweeping.

- We used random horizontal flips during training for CIFAR10; we tried training both with and without flips, and found flips to improve sample quality slightly. We also used random horizontal flips for all other datasets except LSUN Bedroom.

- We tried Adam [31] and RMSProp early on in our experimentation process and chose the former. We left the hyperparameters to their standard values. We set the learning rate to $2 \times 10^{-4}$ without any sweeping, and we lowered it to $2 \times 10^{-5}$ for the $256 \times 256$ images, which seemed unstable to train with the larger learning rate.

- We set the batch size to 128 for CIFAR10 and 64 for larger images. We did not sweep over these values.
- We used EMA on model parameters with a decay factor of 0.9999. We did not sweep over this value.

Final experiments were trained once and evaluated throughout training for sample quality. Sample quality scores and log likelihood are reported on the minimum FID value over the course of training. On CIFAR10, we calculated Inception and FID scores on 50000 samples using the original code from the OpenAI [51] and TTUR [21] repositories, respectively. On LSUN, we calculated FID scores on 50000 samples using code from the StyleGAN2 [30] repository. CIFAR10 and CelebA-HQ were loaded as provided by TensorFlow Datasets (https://www.tensorflow.org/datasets), and LSUN was prepared using code from StyleGAN. Dataset splits (or lack thereof) are standard from the papers that introduced their usage in a generative modeling context. All details can be found in the source code release.

## C   Discussion on related work

Our model architecture, forward process definition, and prior differ from NCSN [55, 56] in subtle but important ways that improve sample quality, and, notably, we directly train our sampler as a latent variable model rather than adding it after training post-hoc. In greater detail:

1. We use a U-Net with self-attention; NCSN uses a RefineNet with dilated convolutions. We condition all layers on $t$ by adding in the Transformer sinusoidal position embedding, rather than only in normalization layers (NCSNv1) or only at the output (v2).

2. Diffusion models scale down the data with each forward process step (by a $\sqrt{1 - \beta_t}$ factor) so that variance does not grow when adding noise, thus providing consistently scaled inputs to the neural net reverse process. NCSN omits this scaling factor.

3. Unlike NCSN, our forward process destroys signal ($D_{\mathrm{KL}}(q(\mathbf{x}_T|\mathbf{x}_0) \,\|\, \mathcal{N}(\mathbf{0}, \mathbf{I})) \approx 0$), ensuring a close match between the prior and aggregate posterior of $\mathbf{x}_T$. Also unlike NCSN, our $\beta_t$ are very small, which ensures that the forward process is reversible by a Markov chain with conditional Gaussians. Both of these factors prevent distribution shift when sampling.

4. Our Langevin-like sampler has coefficients (learning rate, noise scale, etc.) derived rigorously from $\beta_t$ in the forward process. Thus, our training procedure directly trains our sampler to match the data distribution after $T$ steps: it trains the sampler as a latent variable model using variational inference. In contrast, NCSN's sampler coefficients are set by hand post-hoc, and their training procedure is not guaranteed to directly optimize a quality metric of their sampler.

## D   Samples

**Additional samples**   Figure 11, 13, 16, 17, 18, and 19 show uncurated samples from the diffusion models trained on CelebA-HQ, CIFAR10 and LSUN datasets.

**Latent structure and reverse process stochasticity**   During sampling, both the prior $\mathbf{x}_T \sim \mathcal{N}(\mathbf{0}, \mathbf{I})$ and Langevin dynamics are stochastic. To understand the significance of the second source of noise, we sampled multiple images conditioned on the same intermediate latent for the CelebA $256 \times 256$ dataset. Figure 7 shows multiple draws from the reverse process $\mathbf{x}_0 \sim p_\theta(\mathbf{x}_0|\mathbf{x}_t)$ that share the latent $\mathbf{x}_t$ for $t \in \{1000, 750, 500, 250\}$. To accomplish this, we run a single reverse chain from an initial draw from the prior. At the intermediate timesteps, the chain is split to sample multiple images. When the chain is split after the prior draw at $\mathbf{x}_{T=1000}$, the samples differ significantly. However, when the chain is split after more steps, samples share high-level attributes like gender, hair color, eyewear, saturation, pose and facial expression. This indicates that intermediate latents like $\mathbf{x}_{750}$ encode these attributes, despite their imperceptibility.

**Coarse-to-fine interpolation**   Figure 9 shows interpolations between a pair of source CelebA $256 \times 256$ images as we vary the number of diffusion steps prior to latent space interpolation. Increasing the number of diffusion steps destroys more structure in the source images, which the

model completes during the reverse process. This allows us to interpolate at both fine granularities and coarse granularities. In the limiting case of 0 diffusion steps, the interpolation mixes source images in pixel space. On the other hand, after 1000 diffusion steps, source information is lost and interpolations are novel samples.

Figure 9: Coarse-to-fine interpolations that vary the number of diffusion steps prior to latent mixing.

Figure 10: Unconditional CIFAR10 progressive sampling quality over time

Figure 11: CelebA-HQ $256 \times 256$ generated samples

(a) Pixel space nearest neighbors

(b) Inception feature space nearest neighbors

Figure 12: CelebA-HQ $256 \times 256$ nearest neighbors, computed on a $100 \times 100$ crop surrounding the faces. Generated samples are in the leftmost column, and training set nearest neighbors are in the remaining columns.

Figure 13: Unconditional CIFAR10 generated samples

Figure 14: Unconditional CIFAR10 progressive generation

(a) Pixel space nearest neighbors

(b) Inception feature space nearest neighbors

Figure 15: Unconditional CIFAR10 nearest neighbors. Generated samples are in the leftmost column, and training set nearest neighbors are in the remaining columns.

Figure 16: LSUN Church generated samples. FID=7.89

Figure 17: LSUN Bedroom generated samples, large model. FID=4.90

Figure 18: LSUN Bedroom generated samples, small model. FID=6.36

Figure 19: LSUN Cat generated samples. FID=19.75