[Reviews · NeurIPS 2020]

Review 1

Summary and Contributions: This paper takes a closer look at a generative mechanism inspired by out-of-equilibrium thermodynamics and based on "reversing the second law of thermodynamics" proposed a few years ago. The main idea is that one can generate samples from a dataset from noise by thinking of noise samples as the result of a "forward process" that iteratively perturbs the data points. By reversing this forward process we can then recover the data samples from noise. This offers an alternative mechanism to other more established ones like autoregressive flows, GANs, and Variational Autoencoders.

Strengths: - Conceputal novelty: This papers validates a previous idea that seemed intriguing but so far hadn't been able to demonstrate of being able to match the quality of other generative models, in particular for the synthesis of high quality images. This will significantly add to the toolbox of generative models available to the community. - Theoretical advance: The paper establish an interesting connection between generative diffusion models and denoising score matching with Langevin dynamics, which contributes to providing a potentially fruitful conceptual unification within the field. - Methodological advances: The paper proposes assumptions and parametrizations about the probabilistic process and the trained neural network parametrization of the reverse process that are by themselves contributions that might potentially inspire adjacent lines of work.

Weaknesses: The paper doesn't seem to have any glaring weaknesses. The main concern might be tied to the functioning of the generative model, in particular the fact that it relies on sequential multi-step generation with a number of steps that in this particular case is rather high (1000 steps). While this gives some flexibility to the model (as shown by the conditioning experiments at different latent steps), this also raises the question of whether the fact of having to sequentially run hundreds of steps to obtain a sample might not be excessively slow for practical applications. - Providing more details on the neural network parametrization of the reverse process (in particular how position embeddings are being provided) could help reproducibility. On the other hand, provide the code of their implementation, which is as much as one can ask in terms of reproducibility.

Correctness: Everything seems correct. The high quality of the generated samples sort of speaks by itself in regard to the correctness of the method.

Clarity: - The paper is well written. - It could be helpful to provide some more details in regards to some interesting conceptual observations provided by the authors (like the connection to denoising score matching), and the parametrization of the neural network (like how exactly parameters are shared across time step, i.e. which layers of the model receive the sinusoidal position embeddings).

Relation to Prior Work: The relation to previous literature is clearly discussed. In particular, the connection to the unsupervised model based on nonequilibrium thermodynamics and to denosing score matching is clearly mentioned.

Reproducibility: Yes

Additional Feedback: POST-REBUTTAL COMMENTS: I would like to thank the authors for discussing the disadvantages of diffusion models compared to other generative methods, and for the detailed comparison with NCSN. I will maintain my score and commend the authors for a great paper.


Review 2

Summary and Contributions: This paper advances diffusion probabilistic models that succeed to draw high-fidelity samples from high-dimensional data distributions such as the CelebA-HQ (256x256) dataset. The author offers novel insights that, under certain parametrizations (fixed isotropic variance in p_theta), diffusion models are linked to denoising score matching (DSM) with multi-scale noises. Inspired by the DSM, the author also formulated the learning objective to predict the Gaussian noise given the noise-perturbed data at different noise levels. From the empirical results, the quantitative evaluation on CIFAR-10 shows the proposed method achieves new state-of-the-art in Inception and FID score. Qualitative speaking, the author also demonstrates the generalizability of the proposed approach via nearest neighbor results and interpolation experiments.

Strengths: (1) strong empirical results, and thorough experiments in ablation study, distortion rate, interpolation, nearest neighbor. (2) novel insights that connect diffusion models to multi-scale DSM.

Weaknesses: See Additional Feedback part.

Correctness: See Additional Feedback part.

Clarity: The writing is clear and self-contained.

Relation to Prior Work: Author should provide more insights about how the proposed diffusion model differs from NCSN, from both algorithm realization stand point and empirical performance difference perspective.

Reproducibility: Yes

Additional Feedback: From an algorithm realization perspective, the training (Algo. 1) and inference (Algo. 2) of the proposed diffusion model are very similar to the training and inference procedure of NCSN [1,2]. However, the empirical performance of the proposed approach shows huge advantage over NCSN. Can the author elaborate what makes this difference? To my knowledge, the difference are The number of noise-levels (denoted as L): For the diffusion model, L=1000. For NCSNv1, L=10. For NCSNv2, L=232 for CIFAR-10. The scheduling sequence of variance (denoted as beta_t, which is the \sigma^2 in NCSN): For the diffusion model, beta_1=1e-4, beta_T=0.02, and linear schedule is employed. For NCSN, they consider the geometric sequence, and beta_T is much larger for NCSNv2. Architecture parametrization (denoted as \eps_theta, which is the s_\theta in NCSN). The learning rate schedule in inference What are the main reasons that cause the performance gap, making NCSNv1/NCSNv2 inferior to the diffusion model? Regarding the training objective Eq.(14). At line 120-121, The author claimed that it is analogous to the loss weighting used by NCSN objective. However, Eq (14) seems to be the “unweighted” version of NCSN objective, where NCSN will impose \ell(\sigma)=\sigma^2 for each noise-level, but Eq(14) remove that weighting scheme actually. Can you clarify? Finally, there are some experimental details are missing: (1) Model selection: how does the author conduct model selection? Is it like [1,2] using 1k samples to evaluate FID score, and pick the best checkpoint? (2) Model size: how large is the score network? Does the proposed score networks have similar model size compared to [1,2]? (3) Training time: how long does it take to train the model? On the TPU-v3 and modern GPU, respectively? (4) Sampling time: how long does it take for the sampling? What’s the relation between sampling time and data dimensionality? [1] Generative Modeling by Estimating Gradients of the Data distribution. NeurIPS 2019 [2] Improved Techniques for Training Score-based Generative Models. arXiv 2006.09011v1


Review 3

Summary and Contributions: This paper studied diffusion probabilistic models and proposed a method to generate high-quality samples with diffusion models. The authors achieved this by training on a weighted variational bound designed according to the connection between diffusion probabilistic models and denoising score matching with Langevin dynamics. Moreover, they also showed that the proposed model admits a progressive lossy decompression scheme that can be interpreted as a generalization of autoregressive decoding.

Strengths: 1. The claims are well proved, from the aspects of both theoretical grounding and empirical evaluation. Specifically, the quantitative evaluation in Table 1 and visualization results showed that the proposed model can generate high-quality samples and progressive generation. 2. The contribution of this paper is novel. They proposed an effective method to boost the performance of diffusion models and analyzed the proposed model from various aspects. 3. This paper is highly related to the NeurIPS community.

Weaknesses: 1. Motivation: It is not clear what are the advantages and disadvantages of diffusion models compared to GANs. The authors introduced that "Diffusion models are straightforward to define and efficient to train," while did not compare the efficientness of diffusion models and GANs. 2. It would be better to conduct a qualitative comparison of the proposed method and the original diffusion model to show the difference in generation quality. 3. Table 1 shows that the proposed method outperforms other models with a clear margin on the cifar dataset, while Table 3 shows that StyleGAN is better than the proposed model on LSUN datasets. Is the proposed model performs better on generating images with a low resolution?

Correctness: Yes.

Clarity: Yes.

Relation to Prior Work: Yes.

Reproducibility: Yes

Additional Feedback:


Review 4

Summary and Contributions: This paper combines the ideas of diffusion probabilistic models and denoising score matching with Langevin dynamics to implement an autoregressive decoding generative model for image generation. The intuition is clear that diffusion models generally suffer from underfitting while denoising score matching provides better gradients corresponding to gradually decreasing noise levels to improve the learning of the generative model. The experiments carried out on CIFAR10 has demonstrated the effectiveness of the proposed model.

Strengths: Simple idea with reasonably good result

Weaknesses: It is better to have more comparisons to deep generative models with similar hierarchical structures

Correctness: Yes

Clarity: Yes

Relation to Prior Work: Yes

Reproducibility: Yes

Additional Feedback: --Is it possible to apply denoising score matching on other variants of the model instead of diffusion models? For example Laddar VAE where each level draws the sample from a parameterised Gaussian distribution conditioned on a latent variable from a higher level instead of diffusion process. Intuitively the model has better capacity but also suffers from underfitting. --Is the diffusion setup key to the improvement brought by the proposed model --Line 109. "our choice here ensures that the variational bound is a lossless codelength of discrete data". Why is the variational bound a lossless codelength of discrete data? What is the connection between the proposed model and inverse autoregressive flow? --Line 124. "objective (14) is a weighted variational bound that emphasizes different aspects of reconstructions that epsilon_θ must perform". Why is objective (14) a better objective function? What is the intuition behind this? Even if it achieves better performance, why would this reweighting lead to better sample quality?

[Author Response · NeurIPS 2020]

We thank all reviewers for their comments, which overall were positive on novelty, our empirical sample quality results
and ablations, and our connection between diffusion models and denoising score matching (DSM) with Langevin
dynamics. Reviewers generally asked for more discussion on the relationship to other models (e.g. NCSN and GANs).

**R1**: **Slow sampling speed**: this is indeed a disadvantage of diffusion models, just like autoregressive models and score
matching/energy based models with MCMC samplers. We'll discuss this, and we'd like to improve this in future work.

**R2**: **Explanation of empirical advantages over NCSNv1 [2], v2 [3]**: (Note that NCSNv2 [3] appeared on arXiv after
the NeurIPS deadline.) Apart from differences **R2** mentioned, our architecture, forward process definition and prior
are subtle but important choices that improve sample quality, and most importantly, we directly train the sampler
as a latent variable model rather than adding it post-hoc. Details: **(1)** We use a U-Net with self-attention; NCSN
uses a RefineNet with dilated convolutions. We condition all layers on $t$ by adding in the Transformer sinusoidal
position embedding, rather than only in normalization layers (NCSNv1) or only at the output (v2). **(2)** Diffusion
models scale down the data with each forward process step (the $\sqrt{1-\beta_t}$ factor in Eq 2) so that variance does not
grow when adding noise, thus providing consistently scaled inputs to the neural net reverse process. NCSN omits this
scaling factor. **(3)** Unlike NCSN, our forward process destroys signal ($D_{\mathrm{KL}}(q(\mathbf{x}_T|\mathbf{x}_0) \| \mathcal{N}(\mathbf{0},\mathbf{I})) \approx 0$), ensuring a
close match between the prior and aggregate posterior of $\mathbf{x}_T$. Also unlike NCSN, our $\beta_t$ are very small, which ensures
that the forward process is reversible by a Markov chain with conditional Gaussians. Both of these factors prevent
distribution shift when sampling. **(4)** Our Langevin-like sampler (Eq 11, L87) has coefficients derived rigorously
from $\beta_t$ in the forward process. Thus, our training procedure directly trains our sampler to match the data distribution
after $T$ steps: it trains the sampler as a latent variable model using variational inference (see L90-93). In contrast,
NCSN's sampler coefficients are set by hand post-hoc, and their training procedure is not guaranteed to directly
optimize a quality metric of their sampler. **Explanation of loss weighting**: the NCSN loss (Eq 5-6 of [2]), combined
with their choice $\lambda(\sigma_i) = \sigma_i^2$, simplifies to $\frac{1}{L}\sum_{i=1}^{L} \mathbb{E}_{\mathbf{x}}\mathbb{E}_{\boldsymbol{\epsilon}\sim\mathcal{N}(\mathbf{0},\mathbf{I})}\left[\frac{1}{2}\|\sigma_i s_\theta(\mathbf{x}+\sigma_i\boldsymbol{\epsilon},\sigma_i)+\boldsymbol{\epsilon}\|^2\right]$. These MSE terms are
equally weighted, analogous to our "unweighted" Eq 14. NCSNv2 defines $s_\theta(\cdot,\sigma_i) = s_\theta(\cdot)/\sigma_i$, so their loss becomes
$\frac{1}{L}\sum_{i=1}^{L} \mathbb{E}_{\mathbf{x}}\mathbb{E}_{\boldsymbol{\epsilon}\sim\mathcal{N}(\mathbf{0},\mathbf{I})}\left[\frac{1}{2}\|s_\theta(\mathbf{x}+\sigma_i\boldsymbol{\epsilon})+\boldsymbol{\epsilon}\|^2\right]$, which is similar to ours. **Experimental details**: see Appendix B. Like
GAN literature, we picked the best checkpoints according to FID (50k samples on CIFAR10, 2048 on LSUN/CelebA-
HQ). We used 35.7M parameters on CIFAR10, and NCSN used 29.7M. NCSNv2 used 80M-95M parameters for LSUN
($128^2$) and FFHQ ($256^2$); we used 114M for LSUN ($256^2$) and CelebA-HQ ($256^2$). On TPU v3-8 (similar to 8 V100
GPUs), our CIFAR model trains at 21 steps/sec at batch size 128 (10.6 hours to train to completion at 800k steps), and
sampling a batch of 256 images takes 17 sec; our CelebA-HQ/LSUN ($256^2$) models train at 2.2 steps/sec at batch size
64, and sampling a batch of 128 images takes 300 sec. **Sampling time vs. data dimension**: sampling time (Alg. 2)
depends on $T$ and the neural net, which are fixed before training (like how they are fixed before training a hierarchical
VAE). We'd like to investigate how existing MCMC theory on this topic applies to our models.

**R3**: **Performance at high resolution**: since submission, we trained a larger 256M parameter model for $256^2$ LSUN
Bedroom (vs 114M in the submission), **improving FID from 6.36 to 4.90**. We expect more improvements are
possible for high resolutions via model scaling. **GANs**: GANs have fast generation, whereas we used $T = 1000$
steps. Downsides of GANs are training instability, difficulty in capturing the whole data distribution, and difficulty
in evaluating overfitting. In contrast, our model is trained on a simple, stable non-adversarial MSE loss. Like other
likelihood-based models (autoregressive, VAE, flows), our model captures all modes and we can easily check overfitting
by computing test set log likelihood. **Qualitative comparison w/ the original diffusion model**: the baseline (first two
rows in Table 2) is our reimplementation of the original model with a modern neural net; we'll add a qualitative figure.

**R4**: **Comparisons to models with similar hierarchical structures**: the closest is NVAE [4] (appeared on arXiv after
the NeurIPS deadline). NVAE achieves better log likelihoods and has faster generation, but we attain better sample
quality (IS/FID) and provide rate-distortion curves. **DSM on other models**: this is not straightforward because our
equivalence between DSM and the diffusion objective (Eq 8-12) relies on the Gaussian forward process (Eq 4, 6, 7),
which is unique to the diffusion model. However, loss reweighting (Eq 14) could be useful for other models, as shown
in prior work (e.g. beta-VAE, ConvDRAW). "**Is the diffusion setup key to the improvement?**" We believe so: see
the discussion above with **R2**. "**Why is the variational bound a lossless codelength of discrete data?**" Due to the
bits-back argument [1]. We will add details. **Connection to IAF**: we are not aware of a direct connection. Since IAF is
a flow, it preserves information between data and latents, but diffusion models destroy information between $\mathbf{x}_0$ and
$\mathbf{x}_T$ (as we stated in L213-215). **Reweighting and sample quality**: reweighting variational bounds has been shown to
impact sample quality in prior work (e.g. beta-VAE, ConvDRAW). In our case, terms for small $t$ ask the network to
denoise data with very small amounts of noise; since such data is already clean, we down-weight these terms so that the
network can focus on more difficult denoising tasks at larger $t$ terms. We'll add this intuition to the paper.

**[1]** Keeping Neural Networks Simple by Minimizing the Description Length of the Weights (1993) **[2]** Generative Modeling by
Estimating Gradients of the Data Distribution (2019) **[3]** Improved Techniques for Training Score-Based Generative Models (2020)
**[4]** NVAE: A Deep Hierarchical Variational Autoencoder (2020)


[Meta-Review · NeurIPS 2020]

The paper gives insights on DSM (Denoising Score Matching ) and MCMC method and links it to Probabilistic Diffusion models. This is novel and reviewer agrees that the paper has a good contribution. Concerns: • Algorithmically it is the same algorithm of NCSN with 1) different hyper-parameters motivated from diffusion models ( like scaling of inputs between stages ) 2) different architectural choices • The FID is very low , maybe some memorization ? qualitative experiments are done like nearest neighbor and interpolation, can you add FID on a test set not on the training set to measure memorization? Please include in the final version of the paper all the details in answers in rebuttal to R2 on the main comparison with NCSN, architecture choices etc, training time , sampling time, the need for cross-validation etc and how long the full training and cross validation takes. While probabilistic diffusion models are elegant their compute time is intensive please discuss this in the paper, and how you think this can be addressed.